# Design of an Optimal Robust Possibilistic Model in the Distribution Chain Network of Agricultural Products with High Perishability under Uncertainty

**Amir Daneshvar** [1], **Reza Radfar** [2], **Peiman Ghasemi** [3,*], **Mahmonir Bayanati** [4] **and Adel Pourghader Chobar** [5]

1. Department of Information Technology Management, Electronic Branch, Islamic Azad University, Tehran 1477893855, Iran; a_daneshvar@iauec.ac.ir
2. Department of Technology Management, Science and Research Branch, Islamic Azad University, Tehran 1477893855, Iran; r.radfar@srbiau.ac.ir
3. Department of Business Decisions and Analytics, University of Vienna, Kolingasse 14-16, 1090 Vienna, Austria
4. Department of Management, Faculty of Technology and Industrial Management, West Tehran Branch, Islamic Azad University, Tehran 1477893855, Iran; bayanati.mahmonir@wtiau.ac.ir
5. Department of Industrial Engineering, Faculty of Industrial and Mechanical Engineering, Qazvin Branch, Islamic Azad University, Qazvin 3471993116, Iran; apourghader@qiau.ac.ir
* Correspondence: peiman.ghasemi@univie.ac.at

**Abstract:** In this article, the modeling of a distribution network problem of agricultural products with high perishability under uncertainty is discussed. The designed model has three levels of suppliers, distribution centers, and retailers, in which suppliers can directly or indirectly meet retailers' demand. Due to agricultural product distribution network unpredictability, robust possibilistic optimization (RPO) has been applied. This model is innovative and takes uncertainty into account. The findings show that uncertainty increases network demand. Supply, distribution, maintenance, and order expenses have grown. By examining the rate of perishability of agricultural products, it has been revealed that, with the growth of this rate, the costs have increased according to the ordering and spoilage of the products. The genetic algorithm (GA), whale optimization algorithm (WOA), and arithmetic optimization algorithm (AOA) have also been applied to analyze the model. The calculations on 10 sample problems in larger sizes show that the AOA has the best performance in achieving near-optimal solutions. Conversely, the WOA has the lowest computing time compared to other meta-heuristic algorithms. Additionally, the statistical test results show no significant difference between the average calculation time and the objective function among the applied algorithms.

**Keywords:** distribution network of agricultural products; robust possibilistic optimization; perishability; inventory control; meta-heuristic algorithm; uncertainty

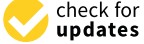



## 1. Introduction

Agriculture is one of the most important and influential sectors in the economy of any country, and it plays an important role in its political and economic independence. Within this, food supply is of particular importance, due to the ever-increasing population growth, because achieving food security is considered one of the main goals of every country. Food security is based on the three principles of the availability and supply of healthy food, the ease of access and purchasing power, and stability in receiving food. The supply chain, or the supply, of basic goods in the agricultural sector includes a set of operations from the farm to the table, which starts with the farmer as a supplier and finally ends with the consumer. In different studies, different steps have been used to explain the supply chain, but, in general, a supply or supply chain includes a set of activities that include the supply of inputs, production, harvesting, storage, processing, distribution, and marketing. The existence of a supply chain for various agricultural products, especially products that have a perishable nature, leads to an increase in food security in a country by significantly

reducing the amount of waste and also reducing the final costs of production. Despite the recent advances in food industry technology and high food and agricultural product output, population expansion and rising food consumption have made food security one of the world's most pressing concerns. The most important sector for providing people with food is the agricultural sector of any country [1]. According to the 2005 International Development Report, agriculture has played a key role in reducing global poverty and increasing food security. Therefore, the food security of about 2.5 billion people in developing countries depends on the agricultural sector [2]. Therefore, the distribution of agricultural products in supply chain networks (SCN)s has been proposed as one of the eight dimensions affecting food security. The SCN of agricultural products starts with their supplier and finally ends with the consumer. In this chain, important issues such as the way to supply agricultural products, the rapid perishability of agricultural products, the amount of product supply, and the way to transport and store agricultural products are discussed [3]. Therefore, one of the biggest problems of the agricultural sector in a country is the lack of awareness among farmers regarding the balanced planting of agricultural products according to the demand and the high perishability of the agricultural products. A disturbance of this balance, on the one hand, has caused an abundance of a product and a significant decrease in its price in one year, causing farmers to suffer, and, conversely, a reduction in other products has caused an increase in the price and dissatisfaction of the people [4].

The key agriculture industry issues are a weak supply chain design and intermediary entrance. Therefore, ignoring these issues leads to the deterioration of perishable products and reduces the direct supply of agricultural and food products to the end consumer [5]. This study provides a distribution network model for agricultural products, focusing on the high perishability of agricultural products from the moment of harvest to the moment of delivery to the end consumer, in a situation where the amount of demand is not precisely known. The distribution network of agricultural products includes a set of suppliers who directly or indirectly (through distribution centers) fulfill the uncertain demands of retailers.

In this network, making important strategic and tactical decisions can help to reduce the costs of the entire distribution network of agricultural products. Among these strategic decisions is the selection of suitable suppliers of agricultural products. Additionally, tactical decisions such as the correct management of the distribution of agricultural products to reduce the length of waiting lines in distribution centers, the optimal routing of product distribution, determining the optimal order quantity of agricultural products according to uncertain demand, and managing the inventory of agricultural products with high perishability can help in network design. This article discusses strategic and tactical choices, transforming the provided model into an NP-hard model with three algorithms, including the genetic algorithm (GA), the whale optimization algorithm (WOA), and the arithmetic optimization algorithm (AOA), and using an RPO approach to manage unknown quantities like anticipated demand and shipping costs.

The paper is structured in such a way that, after the introduction, the second section reviews the relevant literature to clarify the research gap. In the third section, a distribution network model of agricultural products is presented in the conditions of uncertainty and the use of the RPO method to control non-crisp parameters. In the fourth section, the numerical results of the designed model are analyzed, and a sensitivity analysis is performed on different parameters of the problem. Finally, in the fifth section, conclusions and suggestions for future research are discussed.

## 2. Literature Review

Morganti and González-Feliu [6] looked at the food hub in the city of Parma, which is located in Italy, as a case study for their investigation of the urban procurement of perishable goods. Within the scope of this report, they investigated the methods of food distribution utilized by urban distributors such as corporate retail chains, independent stores, hotels, restaurants, and caterers. A model of multi-objective optimization that takes

into account sustainability during the decision-making process was proposed by Govindan et al. [7] for use in an SCN that contains perishable goods. The most important goal on their agenda was to reduce emissions of greenhouse gases, as well as the overall costs. They resolved the issue by utilizing MOPSO in conjunction with a modified version of AMOVNS (multi-objective variable neighborhood search). Accorsi et al. [8] proposed using a linear programming approach to strike a balance between the costs of logistical operations and the emissions of carbon dioxide in an agro-food ecosystem. The results of their model illustrate how environmental resources, production, distribution, and infrastructure are all interconnected with one another. In order to achieve the best possible results with the design of the SCN, De Keizer et al. [9] looked into the structure of the logistics network for perishable goods that had a time of quality decrease. The efficiency with which the network of logistics for fresh agricultural goods operates is significantly influenced both by the amount of time it takes for logistics activities to be completed and by the surrounding environmental factors. As time passes, or the temperature rises, the quality of the product deteriorates, and additional effort is required to deliver the product at an appropriate time and quality level. A multi-period single-objective mathematical model was proposed by Mogale et al. [10] to lower the establishment, maintenance, carbon emission, and risk penalty costs as a result of the increasing levels of hunger all over the world, as well as global food insecurity. According to the findings of the PSO solution applied to the proposed model, the technique that was suggested has high efficiency when it comes to producing the desired outcomes.

In a research study, Huang et al. [11] investigated the optimal pricing strategy and supply chain configuration of perishable foods in an environment of inflation. The supply chain has been severely impacted by the COVID-19 outbreak, which has led to an increase in the price of time-sensitive and perishable goods. They modeled a supply chain for perishable foods, taking into account inflation, and used the discounted cash flow method to calculate profits while inflation was in effect. In order to maximize the efficiency of food distribution and production planning, GÜNER and UTKU [12] developed a model based on mixed-integer programming. In order to find a solution to the problem, they decided to use the CPLEX method as an optimization tool. The results of the case study revealed that the strategy presented in this article may be used for the settlement of issues within a reasonable amount of time. The strategy that was recommended is also applicable to the many different food supply networks. Kara and Dogan [13] investigated the use of learning-based modeling for an inventory of perishable commodities in the presence of stochastic demand and predictable time in order to cut the total costs associated with supply chain operations. The results that were gathered revealed that this model performs better than the other meta-heuristic-based models that are presently being used. Rafie-Majd et al. [14] explored an approach that combined strategic, tactical, and operational optimization in supply chain management with a perishable product in an uncertain demand situation. This method was used to optimize supply chain management with a perishable commodity. They base the optimization approach that they present on the Lagrange method as the basis for the strategy. Gholami-Zanjani et al. [15] established a mathematical model based on a comprehensive two-stage scenario in order to design a food SCN that accommodates fluctuating consumer demand. In order to generate realistic alternatives, they devised a technique based on the Monte Carlo simulation, and to solve the problem, they used Benders' decomposition. Manteghi et al. [16] proposed a model for a sustainable food supply chain as a means of bringing the requirements of the economy and the environment into harmony with one another. The primary goals are to increase profits at SCN and to reduce the amount of emissions of greenhouse gases. In order to accomplish this goal, a variety of competing models were built, and then, using the methods of game theory, the optimal alternatives available were identified for each scenario. Bhat et al. [17] proposed the architecture of agricultural supply chain management by using blockchain and the Internet of Things. This was carried out to overcome the storage and optimization difficulties that are present in agricultural supply chain systems that use a single chain. This design resolves

concerns over scalability, interoperability, security, privacy, and the privacy of linked personal data, as well as concerns over storage. In order to categorize potential security threats, they investigated the several blockchain-based defensive mechanisms that may be accessed in conjunction with (Internet of Things) IoT infrastructure. Khandelwal et al. [18] reviewed 102 scientific publications, including papers presented at conferences and journals, as well as studies and research carried out between the years of 2010 and 2020. This was carried out to identify the problems that are plaguing this industry. In addition to this, the agricultural supply chain management model was inspected and evaluated by them. They said that, despite the growing need for an efficient agricultural supply chain, there are not enough publications that concentrate on the issue as a whole. This is despite the fact that there have been significant increases in demand for such a chain. As a case study of the Colombian agricultural negotiation process, Orjuela et al. [19] proposed the design and development of a platform using a database based on blockchain technologies. The primary objective of this platform was to provide a solution for agricultural supply chain management and control over the Internet. This platform would be designed and developed as part of the Colombian agricultural negotiation process. After agricultural professionals evaluated the blockchain dimensions using the SWARA method, Ronaghi [20], in applied research, devised a method to evaluate blockchain maturity that utilizes each component of blockchain technology and maturity dimensions. After that, the proposed model was put to the test by using the data obtained from a questionnaire that was sent across the supply chain of an organization that is active in the agricultural sector. According to the findings of the research, the three features of blockchain technology that are most crucial are smart contracts, the Internet of Things, and transaction records. Baghizadeh et al. [21] created a multi-objective and multi-product mathematical model that covers economic, social, and environmental objectives in order to build a sustainable supply chain for agricultural goods that are very perishable. They devised the first queuing system to facilitate the movement of harvested items from one structure to another. Emphasizing the fuzzy set theory, the fundamental elements of the problem are treated as if they are unknown, resulting in a powerful hybrid probabilistic programming model. The findings showed that the most effective way to improve all of the target functions was to install drip irrigation systems and solar panels in greenhouses. Together, these two improvements had the greatest impact. A fuzzy mathematical programming model was developed by Babazadeh and Shamsi [22] for the purpose of optimizing the regional wheat center in Iran and achieving sustainable self-sufficiency, in addition to the exchange and export of wheat to countries that are adjacent to Iran, while taking into account uncertainty. They optimized two objective functions (OBFs) by following the approach that was proposed, and the OBFs included both economic and environmental objectives. After the recommended model was tested out in a variety of different kinds of uncertain environments, a probabilistic planning approach was used to deal with the unpredictability of the parameters. The results indicated the effectiveness of the model in directing the wheat supply chain and in making the most optimal strategic and tactical choices. Dündar et al. [23] carried out research to establish a cost-effective network design model, which included transportation and storage, in order to enhance the structure of the wheat supply chain in Turkey. The ability to make decisions at both the tactical and the strategic level is provided by this paradigm. In order to develop the model that was recommended, they used mixed-integer linear programming, and to determine whether or not the model was reliable, it was initially examined with the use of data obtained from 103 interviews with farmers. After that, the model was validated by making use of data collected from the flour milling business via the use of the case study methodology. The desired results were obtained by the use of IBM ILOG CPLEX Optimization Studio 12.6.2.0. Yadav et al. [24] conducted a systematic literature review on the supply chain of agricultural products with the three aims of identifying various challenges in the supply chain of agricultural food, looking into research participation in the field of designing the network of the supply chain of agricultural products, and analyzing the agricultural food supply chain's performance monitoring system by using

various indicators. These goals were accomplished by conducting a review of the supply chain of agricultural products. They looked through 108 different articles to achieve this goal, and, thereafter, they analyzed the most important results, while taking a variety of aspects into account. The research showed that all agricultural stakeholders have, for the most part, come to terms with the digitization of the food supply chain in agriculture. Salehi-Amiri et al. [25] developed a closed-loop SCN for the avocado industry by creating a dual-objective model that took into account the costs of the avocado industry, as well as the social factor of job opportunities, with the two objectives of minimizing the total costs and maximizing employment in various locations. They examined a real-world case study, ran it through the CPLEX solver to see which solutions were the most effective, and then selected the most appropriate locations at which to establish a number of centers in order to put the proposed model to the test. According to the research, this network is the one that is affected by demand the most. Alinezhad et al. [26] investigated the performance of a stable closed-loop SCN that was based on fuzzy theory under uncertain conditions. Their suggested network is a multi-period, multi-product issue that was developed using a two-objective mixed-integer linear programming model with fuzzy demand and rate of return in order to simultaneously optimize the supply chain profit and customer satisfaction. In other words, they want to maximize both of these metrics at the same time. They used fuzzy linear programming and the L-P metric technique, respectively, to cope with the uncertainty that came along with having two different model goals. Mukherjee et al. [27] conducted a study to identify the challenges that are associated with the use of blockchain technology in the food and agriculture supply chain. The obstacles that blockchain technology faces in the food and agricultural supply chain were identified by the authors via the use of several technical, organizational, and environmental frameworks. The information was gathered via the use of a survey, as well as a questionnaire. As examples of empirical techniques, exploratory factor analysis and structural equation modeling were applied. The findings of this study assist service providers in resolving problems that arise when companies use blockchain technology within their enterprises.

Based on the existing literature, the features of the article can be described as follows:

- Designing a model based on the high perishability of agricultural products;
- Considering product distribution queueing;
- The use of the RPO method in controlling non-crisp parameters.

In an innovative way, this research presents a model for optimizing the distribution of agricultural products and highly perishable products.

## 3. Problem Definition and Modeling

In this paper, the design of RPO in the distribution chain network of agricultural products with high perishability has been discussed. Accordingly, the considered distribution network includes suppliers, distribution centers, and retailers. In this model, and according to Figure 1, suppliers send agricultural products to distribution centers for distribution to retailers. It is also possible to send agricultural products directly to retailers in this network. After receiving agricultural products, the distribution centers distribute them to retailers. The distribution of products to retailers by distribution centers is a routing problem.

In this problem, distribution centers and retailers order products from suppliers based on the high perishability of the agricultural products and their reorder points. Due to the uncertain market demand for various products, it is not possible to satisfy all of the customers' demands. Also, each distribution center has a limited number of service providers who can distribute the agricultural products to retailers at different rates.

Therefore, it is possible to create queues in distribution centers. Accordingly, the designed model seeks to balance the length of the queue in order to reduce the waiting time. Thus, a series of strategic and tactical decisions are about minimizing the costs of the entire distribution network, including supplier selection costs, agricultural product transfer costs, product perishability costs, and other operational costs.

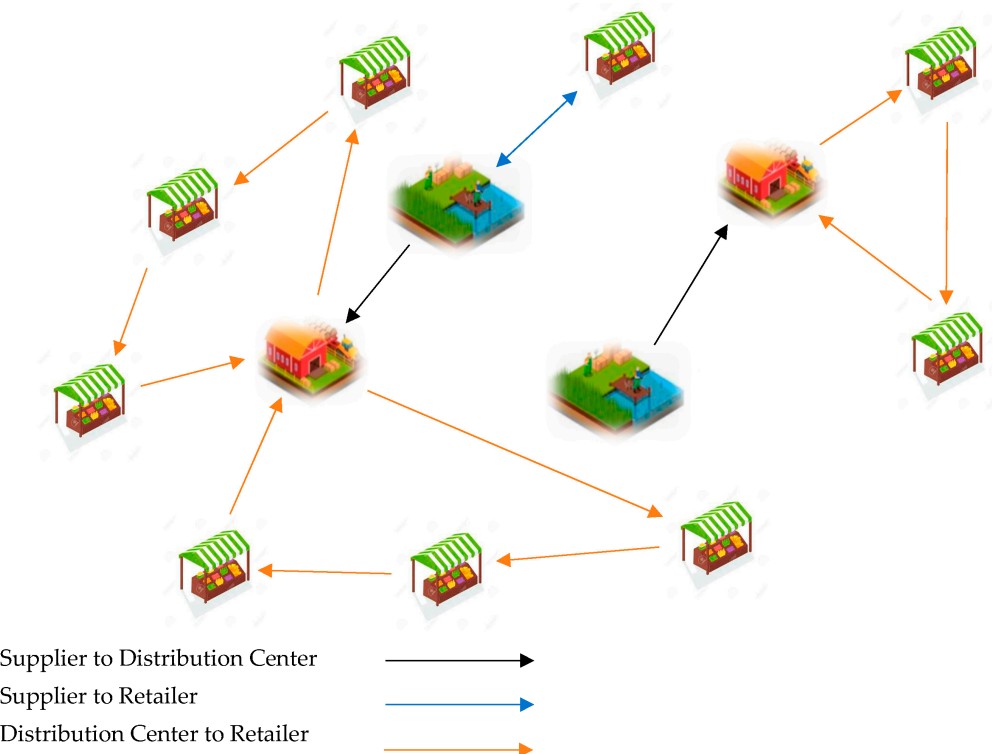

Supplier to Distribution Center

Supplier to Retailer

Distribution Center to Retailer

**Figure 1.** Agricultural products distribution network.

The model in this section is presented according to the following assumptions:

- The presented model is single-period and multi-product;
- Each product has a different perishability rate;
- The demand for perishable products of retailers and transportation costs are uncertain;
- The warehouses of distribution centers and retailers have safety stock;
- The reorder points of distribution centers and retailers are different from each other;
- The number of service providers in the distribution of agricultural products is different in each distribution center;
- The service rate (distribution of products) in distribution centers is fixed and clear.

According to the above assumptions, the mathematical model presented in this section aims to achieve the following: to select suppliers of agricultural products, the optimal allocation of suppliers to distribution centers, the optimal routing of the distribution of agricultural products to retailers, the optimal order quantity of agricultural products with a high perishability rate, and to reduce the length of queues in the distribution of agricultural products in distribution centers.

In the following, the symbols used in the mathematical model are introduced.

Sets:

| | |
|---|---|
| $N$ | Set of retailers $n, l \in N$ |
| $M$ | Set of distribution centers $m, d \in M$ |
| $B$ | Set of potential suppliers $b \in B$ |
| $F$ | Crops $f \in F$ |

Parameters

| | |
|---|---|
| $A_{fm}$ | The cost of ordering product $f \in F$ from distribution center $m \in M$ |
| $H_{fm}$ | The cost of product preservation $f \in F$ in distribution center $m \in M$ |
| $h_{fn}$ | The cost of product preservation $f \in F$ in retailer $n \in N$ |
| $a_{fn}$ | The cost of ordering product $f \in F$ from retailer $n \in N$ |
| $\mu_{fn}$ | Daily demand of product $f \in F$ from retailer $n \in N$ |
| $\sigma_{fn}$ | Standard deviation $\mu_{fn}$ |

| | |
|---|---|
| $\theta_f$ | Perishability rate of product $f \in F$ |
| $\rho_{nl}$ | Correlation coefficient of retailer ($n \in N$) demand and retailer $n \in N$ |
| $l_{fmb}$ | Product ($f \in F$) lead time of distribution center $m \in M$ from supplier $b \in B$ |
| $l_{fnm}$ | Product ($f \in F$) lead time of retailer $n \in N$ from distribution center $m \in M$ |
| $l_{fnb}$ | Product ($f \in F$) lead time of retailer $n \in N$ from supplier $b \in B$ |
| $cap_{fm}$ | Product ($f \in F$) inventory capacity from distribution center $m \in M$ |
| $p_b$ | The fixed cost of supplier ($b \in B$) selection |
| $q_{mb}$ | The fixed cost of building a route between distribution center $m \in M$ and supplier $b \in B$ |
| $t_{fmb}$ | The cost of transporting product ($f \in F$) between distribution center $m \in M$ and supplier $b \in B$ |
| $t_{fnm}$ | The cost of transporting product ($f \in F$) between retailer $n \in N$ and distribution center $m \in M$ |
| $t_{fnb}$ | The cost of transporting product ($f \in F$) between retailer $n \in N$ and supplier $b \in B$ |
| $o_f$ | The cost of perishability of product $f \in F$ |
| $Z_\alpha$ | Cumulative probability distribution function |
| $\mu_m$ | The distribution rate of distribution center $m \in M$ |
| $B_m$ | The upper limit of the queue length for the distribution of agricultural products in distribution center $m \in M$ |
| $\theta_m$ | Upper bound probability for excessive product distribution queue length in distribution center $m \in M$ |
| $\vartheta_m$ | The number of distributors in distribution center $m \in M$ |
| $\varphi_m$ | The cost of waiting time in distribution center $m \in M$ |
| **Decision variables** | |
| $X_{nm}$ | It takes the value 1, if retailer $n \in N$ has received service from distribution centerm $\in M$ |
| $Y_{mb}$ | It takes the value 1, if distribution centerm $\in M$ has received service from supplier $b \in B$ |
| $Z_{nb}$ | It takes the value 1, if retailern $\in N$ has received service from supplierb $\in B$ |
| $W_b$ | It takes the value 1, if the supplier $b \in B$ is selected. |
| $D_{fm}$ | Actual daily demand of product $f \in F$ from distribution center $m \in M$ |
| $U_{fm}$ | Variance of $D_{fm}$ |
| $SS_{fm}$ | Safety stock of product $f \in F$ from distribution center $m \in M$ |
| $SS_{fn}$ | Safety stock of product $f \in F$ from retailer $n \in N$ |
| $R_{fm}$ | Product ($f \in F$) reorder point from distribution center $m \in M$ |
| $R_{fn}$ | Product ($f \in F$) reorder point from retailer $n \in N$ |
| $INV_{fm}$ | Total product ($f \in F$) inventory from distribution center $m \in M$ |
| $INV_{fn}$ | Total product ($f \in F$) inventory from retailer $n \in N$ |
| $Q_{fm}$ | The optimal order point of the product $f \in F$ from the distribution center $m \in M$ |
| $Q_{fn}$ | The optimal order point of the product $f \in F$ from the retailer $n \in N$ |
| $\lambda_m$ | Total products distributed by distribution center $m \in M$ |
| $\pi_{0m}$ | Probability of forming no queue at the distribution center $m \in M$ |
| $T_m$ | The waiting time for the transfer of items in the distribution center $m \in M$ |

According to the defined symbols, the mathematical model of the distribution network of agricultural products with high perishability under the uncertainty of demand and transfer costs is as follows:

$$
\begin{aligned}
Min\ Cost = &\sum_{b \in B} p_b . W_b + \sum_{m \in M} \sum_{b \in B} q_{mb} . Y_{mb} + \sum_{m \in M} \sum_{n \in N} \sum_{f \in F} \frac{\mu_{fn} . X_{nm} . t_{fnm}}{(1-\theta_f)} + \\
&\sum_{m \in M} \sum_{b \in B} \sum_{n \in N} \sum_{f \in F} \frac{\mu_{fn} . X_{nm} . t_{fmb} . Y_{mb}}{(1-\theta_f)^2} + \sum_{b \in B} \sum_{n \in N} \sum_{f \in F} \frac{\mu_{fn} . Z_{nb} . t_{fnb}}{(1-\theta_f)} + \\
&\sum_{m \in M} \sum_{f \in F} H_{fm} . INV_{fm} + \sum_{n \in N} \sum_{f \in F} h_{fn} . INV_{fn} + \sum_{m \in M} \sum_{f \in F} \frac{A_{fm} . D_{fm}}{Q_{fm}} + \\
&\sum_{n \in N} \sum_{f \in F} \frac{a_{fn} . \mu_{fn} . \sum_{b \in B} Z_{nb}}{Q_{fn}(1-\theta_f)} + \sum_{m \in M} \sum_{b \in B} \sum_{n \in N} \sum_{f \in F} \frac{o_f . \theta_f . \mu_{fn} . X_{nm} . Y_{mb}}{(1-\theta_f)^2} + \\
&\sum_{m \in M} \sum_{n \in N} \sum_{f \in F} \frac{o_f . \theta_f . \mu_{fn} . X_{nm}}{(1-\theta_f)} + \sum_{b \in B} \sum_{n \in N} \sum_{f \in F} \frac{o_f . \theta_f . \mu_{fn} . Z_{nb}}{(1-\theta_f)} + \sum_{m \in M} \varphi_m . T_m
\end{aligned}
\tag{1}
$$

$$
s.t. :
$$

$$
Y_{mb} \leq W_b, \ \forall m \in M, b \in B
\tag{2}
$$

$$Z_{nb} \leq W_b, \ \forall n \in N, b \in B \tag{3}$$

$$X_{nm} \leq \sum_{b \in B} Y_{mb}, \ \forall n \in N, m \in M \tag{4}$$

$$\sum_{n \in N} \mu_{fn}.X_{nm} \leq cap_{fm}, \ \forall m \in M, f \in F \tag{5}$$

$$\sum_{n \in N} X_{nm} + \sum_{b \in B} Z_{nb} = 1, \ \forall n \in N \tag{6}$$

$$\sum_{b \in B} Y_{mb} \leq 1, \ \forall m \in M \tag{7}$$

$$Q_{fn} = \sqrt{\left( \frac{2.a_{fn}.\mu_{fn}.\sum_{b \in B} Z_{nb}}{h_{fn}\left(1 - \theta_f\right)} \right)}, \ \forall n \in N, f \in F \tag{8}$$

$$Q_{fm} = \sqrt{\left( \frac{2.A_{fm}.\sum_{n \in N} \mu_{fn}.X_{nm}}{H_{fm}\left(1 - \theta_f\right)} \right)}, \ \forall m \in M, f \in F \tag{9}$$

$$D_{fm} = \frac{\sum_{n \in N} \mu_{fn}.X_{nm}}{1 - \theta_f}, \ \forall m \in M, f \in F \tag{10}$$

$$U_{fm} = \sum_{n \in N} \sum_{l \in N} \rho_{nl}.\sigma_{fn}.\sigma_{fl}.X_{nm}.X_{lm}, \ \forall m \in M, f \in F \tag{11}$$

$$SS_{fm} = Z_\alpha \sqrt{\sum_{n \in N} \sum_{l \in N} \sum_{b \in B} \rho_{nl}.\sigma_{fn}.\sigma_{fl}.X_{nm}.X_{lm}.l_{fmb}.Y_{mb}}, \ \forall m \in M, f \in F \tag{12}$$

$$R_{fm} = Z_\alpha \sqrt{\sum_{n \in N} \sum_{l \in N} \sum_{b \in B} \rho_{nl}.\sigma_{fn}.\sigma_{fl}.X_{nm}.X_{lm}.l_{fmb}.Y_{mb}} \\ + \frac{\sum_{n \in N} \sum_{b \in B} \mu_{fn}.X_{nm}.l_{fmb}.Y_{mb}}{1 - \theta_f}, \ \forall m \in M, f \in F \tag{13}$$

$$INV_{fm} = \frac{Q_{fm}}{2} + SS_{fm}, \ \forall m \in M, f \in F \tag{14}$$

$$SS_{fn} = Z_\alpha.\sigma_{fn}.\sqrt{\sum_{b \in B} l_{fnb}.Z_{nb}}, \ \forall n \in N, f \in F \tag{15}$$

$$INV_{fn} = \frac{\mu_{fn}.\sum_{m \in M} X_{nm}.l_{fnm}}{2} + \frac{Q_{fn}}{2} + Z_\alpha.\sigma_{fn}.\sqrt{\sum_{b \in B} l_{fnb}.Z_{nb}}, \ \forall n \in N, f \in F \tag{16}$$

$$R_{fn} = Z_\alpha.\sigma_{fn}.\sqrt{\sum_{b \in B} l_{fnb}.Z_{nb}} + \sum_{b \in B} \mu_{fn}.l_{fnb}.Z_{nb}, \ \forall n \in N, f \in F \tag{17}$$

$$\lambda_m = \sum_{n \in N} \sum_{f \in F} \frac{\mu_{fn}.X_{nm}.t_{fnm}}{\left(1 - \theta_f\right)}, \ \forall m \in M \tag{18}$$

$$\pi_{0m} = \left[ \sum_{d=0}^{\vartheta_m - 1} \frac{1}{d!} \left( \frac{\lambda_m}{\mu_m} \right)^d + \frac{1}{\vartheta_m!} \left( \frac{\lambda_m}{\mu_m} \right)^{\vartheta_m} \left( \frac{\vartheta_m \mu_m}{\vartheta_m \mu_m - \lambda_m} \right) \right]^{-1}, \ \forall m \in M \tag{19}$$

$$\left( \sum_{d=0}^{\vartheta_m} \frac{(\lambda_m)^d}{j'!(\mu_m)^d} + \sum_{d=\vartheta_m + 1}^{\vartheta_m + B_m} \frac{(\lambda_m)^d (\vartheta_m)^{d - \vartheta_m}}{d!(\mu_m)^d} \right) \pi_{0m} \geq 1 - \theta_m, \ \forall m \in M \tag{20}$$

$$T_m = \left[ \frac{\pi_{0m}}{\vartheta_m!} \left( \frac{\lambda_m}{\mu_m} \right)^{\vartheta_m!} \frac{\vartheta_m \mu_m}{(\vartheta_m \mu_m - \lambda_m)^2} + \frac{1}{\mu_m} \right], \ \forall m \in M \tag{21}$$

$$X_{nm}, Y_{mb}, Z_{nb}, W_b \in \{0,1\} \tag{22}$$

$$D_{fm}, U_{fm}, SS_{fm}, R_{fm}, R_{fn}, INV_{fm}, INV_{fn}, Q_{fm}, Q_{fn}, \lambda_m, \pi_{0m}, T_m \geq 0 \tag{23}$$

The overall expenses of the distribution network for agricultural goods are reduced as much as possible by Equation (1). In this regard, there are costs such as supplier selection, transportation, costs of ordering and preserving perishable products, costs associated with product perishability, and the cost of waiting time for the distribution of products in the queue of distribution centers. All of these costs add up. The answer to this issue may be found in Equation (2), which demonstrates that, if a supplier is chosen, agricultural goods can be moved from that provider to the distribution centers. Equation (3) demonstrates that, if a supplier is chosen, agricultural goods may be supplied straight from that provider to retail outlets. Equation (4) ensures that, if the distribution center has already sent the agricultural goods to the retailers, then it must have already received the items from the supplier. This is the case even if the distribution center has not yet given the products to the retailers. The solution to Equation (5) ensures that the distribution center will not distribute more items than are physically possible, given its capacity. Equation (6) ensures that only a single distribution center can serve a given retailer's needs for product delivery. Equation (7) ensures that each distribution center will only be able to receive goods and services from a single provider of those goods and services. The ideal order quantity of perishable items for both the distribution center and the store may be found by solving Equations (8) and (9).

Equations (10) and (11) calculate the demand for perishable products. In these relationships, the daily demand for products for retailers follows a normal distribution function in the form of $(\mu_{fn}, \sigma_{fn})$. Therefore, according to the interactions between retailers, the product demand in the distribution center will follow a multivariate normal distribution (D_fm, U_fm). Equations (12) and (13) calculate the safety stock and re-order point for a distribution center. Equation (14) shows the total inventory of the distribution center, including the safety stock and the vendor inventory. Equation (15) shows the retailer's safety stock and Equation (16) shows the retailer's total inventory. Equation (17) shows the retailer's reorder point.

Equation (18) calculates the total amount of agricultural products distributed by each distribution center. Equation (19) shows the probability that there is no distribution queue in the distribution centers. Equation (20) limits the maximum queue length allowed in each distribution center. Equation (21) shows the waiting time for product distribution in each distribution center. Equations (22) and (23) show the types of decision variables.

A RPO technique has been applied to manage the uncertainty of anticipated demand and transfer costs, which may impact crucial distribution network choices for agricultural goods. First, an RPO technique was described, then its use in agricultural produce distribution was examined.

*RPO Approach*

Consider the following model:

$$min \ Cost = fY + \tilde{c}X$$
$$s.t. :$$
$$aX \geq \tilde{d}$$
$$eX \leq sY \tag{24}$$
$$Y \in \{0,1\}, \ X \geq 0$$

where the v$\widetilde{d}$, $\widetilde{c}$, $f$, and $s$ reflect, respectively, the fixed construction cost, the variable cost (transportation), the quantity of demand, and the facility capacity. In addition, the matrix of coefficients is denoted by the letters $a$ and $e$, and both $X$ and $Y$ are continuous variables, with the values zero and one, respectively.

$$min\ Cost = E[Z] = fY + E[\widetilde{c}]X$$
$$s.t. :$$
$$NEC\left\{aX \geq \widetilde{d}\right\} \geq \alpha \qquad (25)$$
$$eX \leq sY$$
$$Y \in \{0,1\},\ X \geq 0$$

where $\alpha$ regulates the minimal level of confidence when using a (pessimistic) decision-making strategy to construct the non-crisp constraint. The generic form of Equation (25) may be derived from the trapezoidal membership functions for fuzzy parameters as follows:

$$min\ Cost = fY + \left(\frac{c^1 + c^2 + c^3 + c^4}{4}\right)X$$
$$s.t. :$$
$$aX \geq \left[(1 - \alpha)d^3 + \alpha d^4\right] - \left[\left(h^m + \frac{v_t + v_t'}{4}\right)(1 - \varepsilon)\right] \qquad (26)$$
$$eX \leq sY$$
$$Y \in \{0,1\},\ X \geq 0$$

In the relations that have been shown so far, the symbol indicates the least amount of certainty in the value of the fuzzy levels of numbers. In addition, the degree to which the constraints may be relaxed is determined by a factor called $\left[\left(\frac{v_t + v_t'}{4}\right)(1 - \varepsilon)\right]$, which represents the many ways in which the flexible constraints can be broken. The parameters $v_t$ and $v_t'$ describe the distance between the lateral boundaries of the unknown parameter and its most probable value. This distance may be thought of as the parameter's "range". They alert you to possible violations of soft limitations, which are explained as follows:

$$v_t = d^4 - d^3$$
$$v_t' = d^2 - d^1$$
$$h^m = \frac{d^2 + d^1}{2} \qquad (27)$$

It is important to note that the parameters display the minimal level of flexibility with limits, whose value ought to be decided by the knowledge and expertise of specialists in the area ($0 \leq \varepsilon \leq 1$). The created model performs better in terms of risk aversion, as soft constraints are more fully satisfied. It is clear that it is challenging for decision makers to reach the ideal satisfaction levels, and adding additional ambiguous metrics and soft limitations might make the process more challenging and time-consuming. Furthermore, since the OBF is formulated based on the expected value of the fuzzy numbers, any departure from the uncertain parameters can result in the unreliability of the results obtained from the model and cause significant losses to businesses. In addition, there is no assurance that the final value is the optimum degree of fulfillment, nor that it can bring about the optimum value for the purpose. The controlled model, which was discussed before, may be understood in this context by considering the following:

$$min\ Cost = fY + \left(\frac{c^1 + c^2 + c^3 + c^4}{4}\right)x + \eta\left[d^4 - (1-\alpha)d^3 - \alpha d^4\right]$$

$$+\varrho\left[\left(h^m + \frac{v_t + v'_t}{4}\right)(1-\varepsilon)\right]$$

$$s.t.:$$

$$aX \geq \left[(1-\alpha)d^3 + \alpha d^4\right] - \left[\left(h^m + \frac{v_t + v'_t}{4}\right)(1-\varepsilon)\right]$$

$$eX \leq sY$$

$$Y \in \{0,1\},\ X \geq 0$$

(28)

In the above relation, $\eta$ and $\varrho$ show the penalty cost of the demand deviation and the penalty weight coefficient of the OBF, respectively. As a result, the controlled model with the RPO method for the distribution network of agricultural products is as follows:

$$Min\ Cost = \sum_{b \in B} p_b.W_b + \sum_{m \in M}\sum_{b \in B} q_{mb}.Y_{mb} + \sum_{m \in M}\sum_{n \in N}\sum_{f \in F}\frac{G_{fn}.X_{nm}.t_{fnm}}{\left(1-\theta_f\right)} +$$

$$\sum_{m \in M}\sum_{b \in B}\sum_{n \in N}\sum_{f \in F}\frac{G_{fn}.X_{nm}.t_{fmb}.Y_{mb}}{\left(1-\theta_f\right)^2} + \sum_{b \in B}\sum_{n \in N}\sum_{f \in F}\frac{G_{fn}.Z_{nb}.t_{fnb}}{\left(1-\theta_f\right)} +$$

$$\sum_{m \in M}\sum_{f \in F} H_{fm}.INV_{fm} + \sum_{n \in N}\sum_{f \in F} h_{fn}.INV_{fn} + \sum_{m \in M}\sum_{f \in F}\frac{A_{fm}.D_{fm}}{Q_{fm}} +$$

$$\sum_{n \in N}\sum_{f \in F}\frac{a_{fn}.G_{fn}.\sum_{b \in B} Z_{nb}}{Q_{fn}\left(1-\theta_f\right)} + \sum_{m \in M}\sum_{b \in B}\sum_{n \in N}\sum_{f \in F}\frac{o_f.\theta_f.G_{fn}.X_{nm}.Y_{mb}}{\left(1-\theta_f\right)^2} +$$

$$\sum_{m \in M}\sum_{n \in N}\sum_{f \in F}\frac{o_f.\theta_f.G_{fn}.X_{nm}}{\left(1-\theta_f\right)} + \sum_{b \in B}\sum_{n \in N}\sum_{f \in F}\frac{o_f.\theta_f.G_{fn}.Z_{nb}}{\left(1-\theta_f\right)} + \sum_{m \in M}\varphi_m.T_m +$$

$$\eta\sum_{n \in N}\sum_{f \in F}\left(\mu_{fn}^4 - (1-\alpha)\mu_{fn}^3 - \alpha\mu_{fn}^4\right) +$$

$$\varrho\sum_{n \in N}\sum_{f \in F}\left(\left[\left(\left(\frac{\mu_{fn}^1 + \mu_{fn}^2}{2}\right) + \left(\frac{\mu_{fn}^4 - \mu_{fn}^3 + \mu_{fn}^2 - \mu_{fn}^1}{4}\right)\right)(1-\varepsilon)\right]\right)$$

(29)

$$s.t.:$$

$$Y_{mb} \leq W_b,\ \forall m \in M, b \in B$$

(30)

$$Z_{nb} \leq W_b,\ \forall n \in N, b \in B$$

(31)

$$X_{nm} \leq \sum_{b \in B} Y_{mb},\ \forall n \in N, m \in M$$

(32)

$$\sum_{n \in N} G_{fn}.X_{nm} \leq cap_{fm},\ \forall m \in M, f \in F$$

(33)

$$\sum_{n \in N} X_{nm} + \sum_{b \in B} Z_{nb} = 1,\ \forall n \in N$$

(34)

$$\sum_{b \in B} Y_{mb} \leq 1,\ \forall m \in M$$

(35)

$$Q_{fn} = \sqrt{\left(\frac{2.a_{fn}.G_{fn}.\sum_{b \in B} Z_{nb}}{h_{fn}\left(1-\theta_f\right)}\right)},\ \forall n \in N, f \in F$$

(36)

$$Q_{fm} = \sqrt{\left( \frac{2.A_{fm}.\sum_{n \in N} G_{fn}.X_{nm}}{H_{fm}\left(1 - \theta_f\right)} \right)}, \ \forall m \in M, f \in F \tag{37}$$

$$D_{fm} = \frac{\sum_{n \in N} G_{fn}.X_{nm}}{1 - \theta_f}, \ \forall m \in M, f \in F \tag{38}$$

$$U_{fm} = \sum_{n \in N} \sum_{l \in N} \rho_{nl}.\sigma_{fn}.\sigma_{fl}.X_{nm}.X_{lm}, \ \forall m \in M, f \in F \tag{39}$$

$$SS_{fm} = Z_\alpha \sqrt{\sum_{n \in N} \sum_{l \in N} \sum_{b \in B} \rho_{nl}.\sigma_{fn}.\sigma_{fl}.X_{nm}.X_{lm}.l_{fmb}.Y_{mb}}, \ \forall m \in M, f \in F \tag{40}$$

$$R_{fm} = Z_\alpha \sqrt{\sum_{n \in N} \sum_{l \in N} \sum_{b \in B} \rho_{nl}.\sigma_{fn}.\sigma_{fl}.X_{nm}.X_{lm}.l_{fmb}.Y_{mb}} \\ + \frac{\sum_{n \in N} \sum_{b \in B} G_{fn}.X_{nm}.l_{fmb}.Y_{mb}}{1 - \theta_f}, \ \forall m \in M, f \in F \tag{41}$$

$$INV_{fm} = \frac{Q_{fm}}{2} + SS_{fm}, \ \forall m \in M, f \in F \tag{42}$$

$$SS_{fn} = Z_\alpha.\sigma_{fn}.\sqrt{\sum_{b \in B} l_{fnb}.Z_{nb}}, \ \forall n \in N, f \in F \tag{43}$$

$$INV_{fn} = \frac{G_{fn}.\sum_{m \in M} X_{nm}.l_{fnm}}{2} + \frac{Q_{fn}}{2} + Z_\alpha.\sigma_{fn}.\sqrt{\sum_{b \in B} l_{fnb}.Z_{nb}}, \ \forall n \in N, f \in F \tag{44}$$

$$R_{fn} = Z_\alpha.\sigma_{fn}.\sqrt{\sum_{b \in B} l_{fnb}.Z_{nb}} + \sum_{b \in B} G_{fn}.l_{fnb}.Z_{nb}, \ \forall n \in N, f \in F \tag{45}$$

$$\lambda_m = \sum_{n \in N} \sum_{f \in F} \frac{G_{fn}.X_{nm}.t_{fnm}}{\left(1 - \theta_f\right)}, \ \forall m \in M \tag{46}$$

$$\pi_{0m} = \left[ \sum_{d=0}^{\vartheta_m - 1} \frac{1}{d!}\left(\frac{\lambda_m}{\mu_m}\right)^d + \frac{1}{\vartheta_m!}\left(\frac{\lambda_m}{\mu_m}\right)^{\vartheta_m}\left(\frac{\vartheta_m \mu_m}{\vartheta_m \mu_m - \lambda_m}\right) \right]^{-1}, \ \forall m \in M \tag{47}$$

$$\left( \sum_{d=0}^{\vartheta_m} \frac{(\lambda_m)^d}{j'!(\mu_m)^d} + \sum_{d=\vartheta_m+1}^{\vartheta_m + B_m} \frac{(\lambda_m)^d (\vartheta_m)^{d-\vartheta_m}}{d!(\mu_m)^d} \right) \pi_{0m} \geq 1 - \theta_m, \ \forall m \in M \tag{48}$$

$$T_m = \left[ \frac{\pi_{0m}}{\vartheta_m!}\left(\frac{\lambda_m}{\mu_m}\right)^{\vartheta_m!} \frac{\vartheta_m \mu_m}{(\vartheta_m \mu_m - \lambda_m)^2} + \frac{1}{\mu_m} \right], \ \forall m \in M \tag{49}$$

$$G_{fn} = \left[ \alpha \mu_{fn}^4 + (1-\alpha)\mu_{fn}^3 \right] \\ + \left[ \left( \left( \frac{\mu_{fn}^1 + \mu_{fn}^2}{2} \right) + \left( \frac{\mu_{fn}^4 - \mu_{fn}^3 + \mu_{fn}^2 - \mu_{fn}^1}{4} \right) \right)(1 - \varepsilon) \right] \tag{50}$$

$$X_{nm}, Y_{mb}, Z_{nb}, W_b \in \{0, 1\} \tag{51}$$

$$D_{fm}, U_{fm}, SS_{fm}, R_{fm}, R_{fn}, INV_{fm}, INV_{fn}, Q_{fm}, Q_{fn}, \lambda_m, \pi_{0m}, T_m, G_{fn} \geq 0 \tag{52}$$

The model presented in this section is a mixed-integer non-linear programming model, which is considered NP-hard due to the integration of strategic and tactical decisions. Therefore, meta-heuristic algorithms have been used to solve the problem and analyze the results. The most important part of the implementation of any algorithm is the appropriate

definition of the initial solution to solve the problem. Each of the algorithms is suitable for its operators to improve the initial solution and achieve the best solution.

## 4. Analysis of the Results

The proposed problem has been analyzed using three different algorithms, including the genetic algorithm (GA), the whale optimization algorithm (WOA), and the arithmetic optimization algorithm (AOA). Because we did not have access to data from the real world, we utilized fictitious information instead, which was generated using the uniform distribution function shown in Table 1.

**Table 1.** Range limits of problem parameters.

| Parameter | Range Limits | Parameter | Range Limits |
|---|---|---|---|
| $A_{fm}$ | $\sim U(20\text{–}40)$ | $\sigma_{fn}$ | $\sim U(50\text{–}70)$ |
| $H_{fm}$ | $\sim U(10\text{–}12)$ | $\theta_f$ | $\sim U(0.6\text{–}0.8)$ |
| $h_{fn}$ | $\sim U(8\text{–}10)$ | $\rho_{nl}$ | $\sim U(0.8\text{–}1.2)$ |
| $a_{fn}$ | $\sim U(25\text{–}45)$ | $l_{fmb}$ | $\sim U(20\text{–}40)$ |
| $\varphi_m$ | $\sim U(20\text{–}30)$ | $l_{fnm}$ | $\sim U(20\text{–}40)$ |
| $p_b$ | $\sim U(8000\text{–}10,000)$ | $l_{fnb}$ | $\sim U(20\text{–}40)$ |
| $q_{mb}$ | $\sim U(800\text{–}1000)$ | $cap_{fm}$ | $\sim U(2000\text{–}2500)$ |
| $B_m$ | $\sim U(8\text{–}10)$ | $o_f$ | $\sim U(100\text{–}120)$ |
| $\theta_m$ | 0.6 | $Z_\alpha$ | 1.96 |
| $\vartheta_m$ | $\sim U(5\text{–}7)$ | $\mu_m$ | $\sim U(800\text{–}1000)$ |
| $\mu_{fn}$ | $\sim U[(400\text{–}500), (500\text{–}600), (600\text{–}700), (700\text{–}800)]$ | | |
| $t_{fmb}$ | $\sim U[(20\text{–}30), (30\text{–}40), (40\text{–}50), (50\text{–}60)]$ | | |
| $t_{fnm}$ | $\sim U[(20\text{–}30), (30\text{–}40), (40\text{–}50), (50\text{–}60)]$ | | |
| $t_{fnb}$ | $\sim U[(20\text{–}30), (30\text{–}40), (40\text{–}50), (50\text{–}60)]$ | | |

To analyze the results and the sensitivity analysis, a numerical example with 10 retailers, 3 distribution centers, 3 suppliers, and 3 types of agricultural products is considered. Therefore, before presenting the results, in Table 2, the suggested levels of the parameters of meta-heuristic algorithms to be adjusted with the Taguchi method are presented.

**Table 2.** Suggested levels of primary parameters.

| Approach | Factor | L1 | L2 | L3 |
|---|---|---|---|---|
| GA | Max it | 100 | 200 | 300 |
| | N pop | 100 | 150 | 200 |
| | Pm | 0.02 | 0.04 | 0.06 |
| | Pc | 0.7 | 0.8 | 0.9 |
| WOA | Max it | 100 | 200 | 300 |
| | N whale | 100 | 150 | 200 |
| | b | 1 | 1.2 | 1.5 |
| AOA | Max it | 100 | 200 | 300 |
| | N pop | 100 | 150 | 200 |
| | $\alpha$ | 3 | 5 | 7 |
| | $\mu$ | 0.2 | 0.3 | 0.5 |

Based on Taguchi's method, three different levels are proposed for each algorithm. Therefore, each of the algorithms is implemented based on Taguchi's tests, and, based on Equation (53), their RPD value is calculated in order to determine the optimal level.

$$RPD_i = \frac{AlgSol_i - BestSol}{BestSol} \tag{53}$$

In the above relation, $AlgSol_i$ is the OBF and $BestSol$ is the OBF. After performing various tests with the Taguchi method, the average graph of the S/N ratio for each algorithm has been obtained, as described in Figure 2.

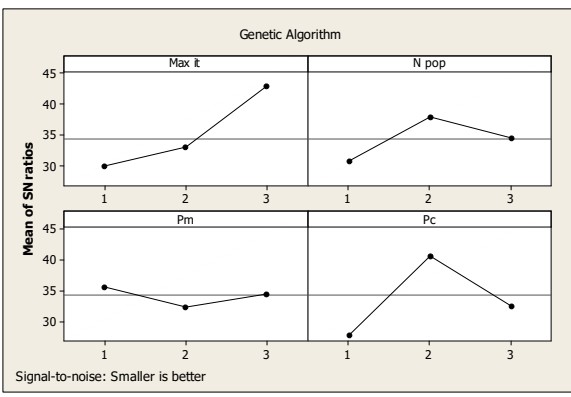

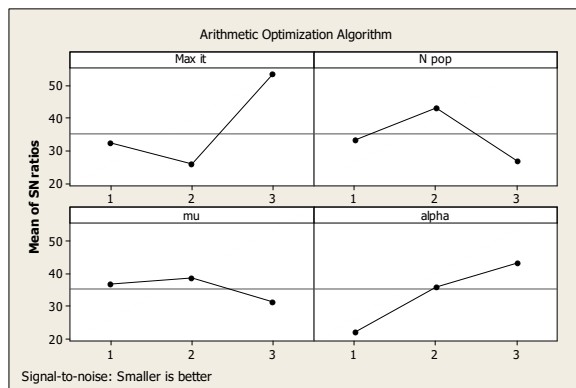

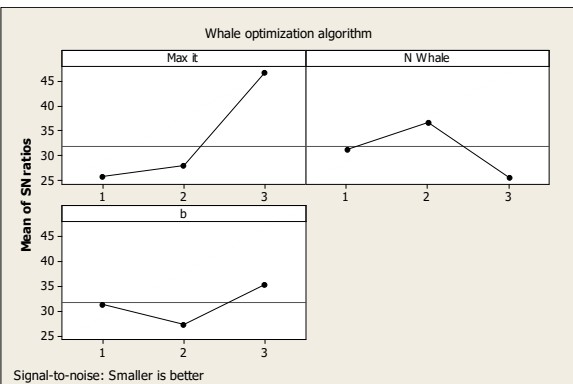

| Algorithm | Parameter | Optimum Value |
|-----------|-----------|---------------|
| GA | Max it | 300 |
| | N pop | 150 |
| | Pm | 0.02 |
| | Pc | 0.8 |
| WOA | Max it | 300 |
| | N whale | 150 |
| | b | 1.5 |
| AOA | Max it | 300 |
| | N pop | 150 |
| | $\alpha$ | 7 |
| | $\mu$ | 0.3 |

**Figure 2.** Average graphs of S/N ratio for meta-heuristic algorithms.

After tweaking the parameters and identifying the ideal levels, the sample issue developed using the suggested techniques was solved, and the convergence of the algorithms in obtaining the optimal solution in 300 iterations is shown in Figure 3.

After solving the sample issue, the GA had an optimum OBF of 18,245,332.24, the WOA was 18,124,774.30, and the AOA was 18,103,488.17. Thus, the highest relative difference between the outcomes was less than 1%, demonstrating the algorithms' effectiveness in finding the near-optimal solution. As shown in Table 1, the problem's key decision variables, including routing and allocation, have been acquired (3).

From the results shown in Table 3, it can be seen that the demand of all of the retailers has been met directly or indirectly by the distribution centers and suppliers. It is only in the WOA that the agricultural products are not sent directly from the supplier to the retailer. Several sensitivity analyses have been performed to determine how critical issue factors affect the OBF value.

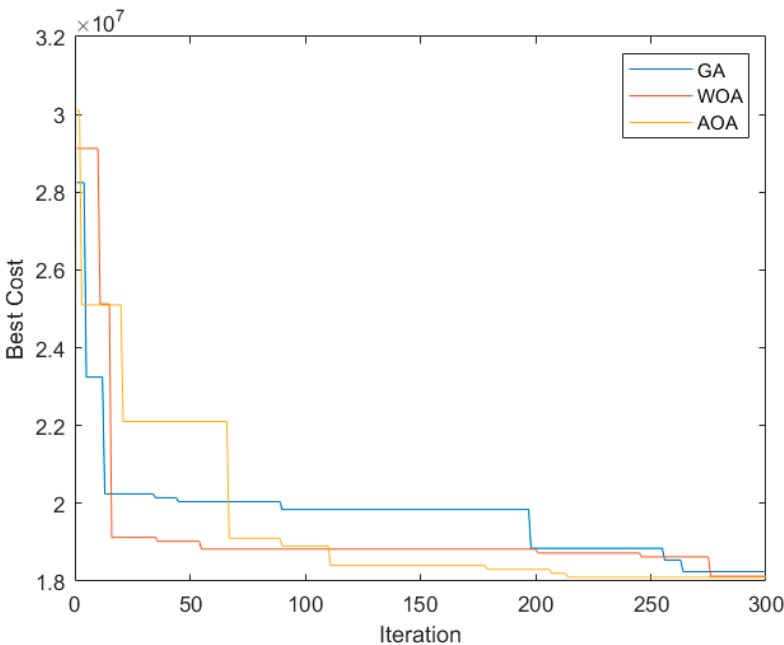

**Figure 3.** Meta-heuristic algorithms converge in the OBF optimization.

**Table 3.** Routing and optimal allocation obtained from solving the numerical example.

| Algorithm | DC to Retailer | Supplier to DC | Supplier to Retailer |
|---|---|---|---|
| GA | $m1 \to n2 \to n3 \to n4 \to m1$<br>$m2 \to n5 \to n10 \to n6 \to m2$<br>$m3 \to n7 \to n1 \to n8 \to n3$ | $b1 \to m3$<br>$b2 \to m2$<br>$b3 \to m1$ | $b3 \to n1$ |
| WOA | $m1 \to n4 \to n1 \to n10 \to n6 \to m1$<br>$m2 \to n3 \to n7 \to n2 \to m2$<br>$m3 \to n9 \to n5 \to n8 \to n3$ | $b1 \to m3$<br>$b2 \to m2$<br>$b3 \to m1$ | |
| AOA | $m1 \to n5 \to n8 \to n9 \to m1$<br>$m2 \to n1 \to n3 \to m2$<br>$m3 \to n7 \to n10 \to n4 \to n2 \to n3$ | $b1 \to m3$<br>$b2 \to m2$<br>$b3 \to m1$ | $b2 \to n6$ |

Table 4 illustrates the OBF from the AOA, which is more efficient than the other methods in different uncertainty rates.

**Table 4.** The total cost of the distribution network of agricultural products in various rates of uncertainty.

| Uncertainty Rate | Total Cost | Percentage of Changes |
|---|---|---|
| 0.1 | 17,432,599.41 | −3.706 |
| 0.2 | 17,568,522.34 | −2.955 |
| 0.3 | 17,732,247.67 | −2.051 |
| 0.4 | 17,934,776.28 | −0.932 |
| 0.5 | 18,103,488.17 | 0.000 |
| 0.6 | 18,324,785.34 | 1.222 |
| 0.7 | 18,568,745.67 | 2.570 |
| 0.8 | 18,864,845.64 | 4.206 |
| 0.9 | 19,146,543.17 | 5.762 |

An uncertainty rate of 0.5 was applied to the numerical example at the beginning. Table 4 shows that the retailers and distribution facilities have more optimum economic order when uncertainty rises, owing to future demand. In addition, the safety stock and the stock at the end of the period of retailers and distribution centers have also increased. This growth in quantity has led to an increase in the costs of the distribution network of agricultural products. Therefore, in the most pessimistic case, and at an uncertainty rate of 0.9, the costs of the entire network have increased by 5.762% compared to the base case, with an uncertainty rate of 0.5. Figure 4 shows the changing trend of the total cost of the distribution network of agricultural products at various rates of uncertainty.

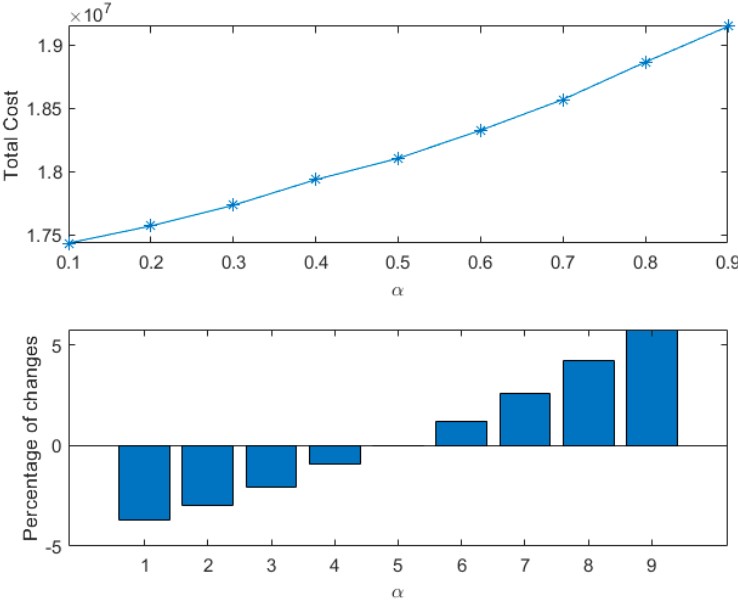

**Figure 4.** The general pattern of fluctuating total costs across different levels of uncertainty in the agricultural goods distribution network.

The distribution network model of agricultural products presented in this article is based on products that perish at a high rate. Due to the rapid consumption of agricultural products, the optimal economic order amount and the reserve stock should be calculated in such a way as to prevent the spoilage and destruction of the product. Therefore, in Table 5, the changes in the total cost of the distribution network of agricultural products at different rates of perishability have been investigated. Also, Figure 4 shows the changing trend of the total cost of the distribution network of agricultural products with different rates of perishability.

**Table 5.** Total cost of distribution network of agricultural products with different rates of perishability.

| Perishable Rate | Total Cost | Percentage of Changes |
|---|---|---|
| 0.1 | 16,020,913.8 | −11.504 |
| 0.2 | 16,583,472.3 | −8.396 |
| 0.3 | 17,034,495.2 | −5.905 |
| 0.4 | 17,469,183.2 | −3.504 |
| 0.5 | 17,867,783.3 | −1.302 |
| 0.6 | 18,103,488.2 | 0.000 |
| 0.7 | 18,725,663.2 | 3.437 |
| 0.8 | 19,425,887.5 | 7.305 |
| 0.9 | 20,147,776.4 | 11.292 |

Table 5 shows that transportation, storage, and distribution costs have risen, owing to the increase in perishability, inventory levels, and the ideal economic order quantity to prevent rapid spoiling. Perishability has significantly raised product failure costs.

Based on the results obtained from the analyses performed, and according to Table 5 and Figure 5, it can be seen that, with the increase in the rate of corruption, the total costs of the distribution network of agricultural products have increased, and, at a perishability rate of 90%, the total costs of the network have increased by 11.292%. Finally, the RPO method's effect on uncertainty parameters compared to the crisp state was examined, and the cost changes in the distribution network of highly perishable agricultural items are displayed in Table 6.

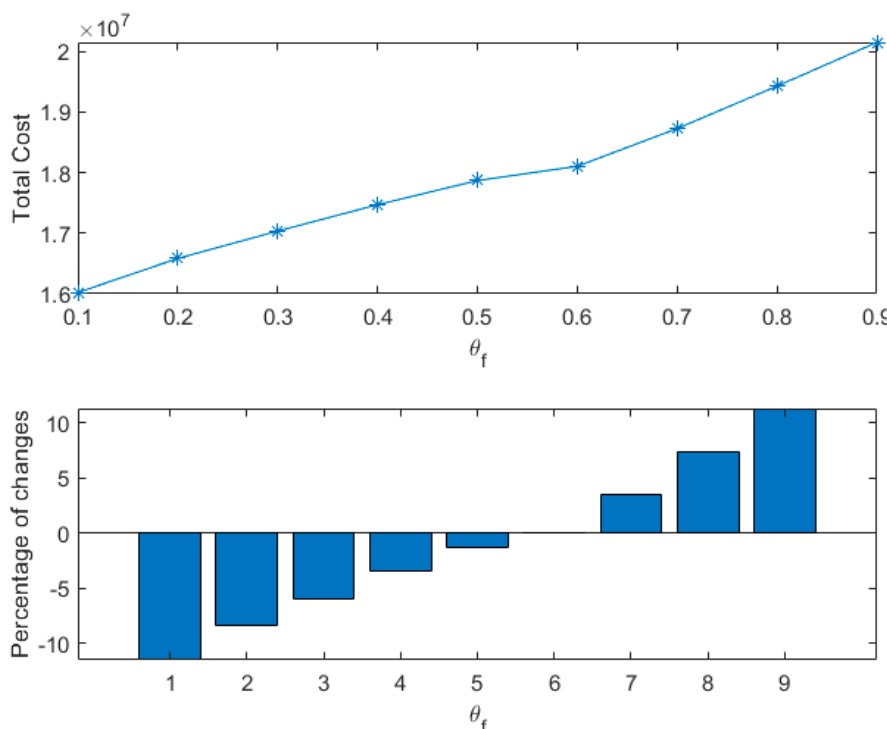

**Figure 5.** The trend of changes in the total cost of the distribution network of agricultural products with different rates of perishability.

**Table 6.** Changes in the total cost of the distribution network of agricultural products and the standard deviation of the changes in the robust state.

| Scenario | $\eta$ | $\varrho$ | Total Cost (Robustness) | Total Cost (Crisp) | Standard Deviation |
|---|---|---|---|---|---|
| 1 | 1 | 1 | 17,268,841.25 | | 63,948.91 |
| 2 | 1 | 2 | 17,864,945.64 | | 66,749.17 |
| 3 | 1 | 3 | 18,097,163.34 | | 71,193.45 |
| 4 | 2 | 1 | 17,632,456.22 | | 59,876.23 |
| 5 | 2 | 2 | 18,103,488.17 | 17,032,648.34 | 61,557.18 |
| 6 | 2 | 3 | 18,565,792.30 | | 69,491.27 |
| 7 | 3 | 1 | 18,223,458.47 | | 48,264.22 |
| 8 | 3 | 2 | 18,647,945.43 | | 53,487.66 |
| 9 | 3 | 3 | 19,219,453.47 | | 57,944.51 |

This research models uncertainty parameters as trapezoidal fuzzy numbers and considers the uncertainty rate. Table 6 shows the changes in the RPO method's total cost compared to the crisp state and the standard deviation of the changes.

Figure 5 shows that the resilience of the agricultural product distribution network model increases the overall expenses while decreasing the standard deviation of the results from running the model five times. This shows that, unlike the increase in the costs of the distribution network of agricultural products compared to the crisp state, the standard deviation of the results has been placed at a lower level, and the assurance of the justification of the problem and not facing a shortage of goods has been guaranteed.

After analyzing the numerical example in small sizes with GA, WOA, and AOA, numerical examples of different sizes have been solved in order to check the efficiency of the algorithms. The results of the AOA in a small numerical example showed that this algorithm is more efficient than the other algorithms. Table 7 shows several numerical examples of larger sizes.

**Table 7.** Numerical examples of larger sizes.

| Problem | *N* | *M* | *B* | *F* | Problem | *N* | *M* | *B* | *F* |
|---------|-----|-----|-----|-----|---------|-----|-----|-----|-----|
| 1 | 12 | 6 | 4 | 4 | 6 | 30 | 10 | 6 | 6 |
| 2 | 15 | 6 | 4 | 4 | 7 | 35 | 12 | 8 | 8 |
| 3 | 18 | 8 | 4 | 5 | 8 | 40 | 12 | 8 | 8 |
| 4 | 20 | 8 | 6 | 5 | 9 | 45 | 15 | 10 | 10 |
| 5 | 25 | 10 | 6 | 6 | 10 | 50 | 18 | 15 | 10 |

Each numerical example presented in Table 7 was executed five times by the GA, the WOA, and the AOA, and the average total costs obtained, as well as the computing time, are shown in Table 8.

**Table 8.** The average costs of the entire distribution network and computing time obtained by meta-heuristic algorithms.

| Problem | Total Cost | | | CPU-Time | | |
|---------|------|------|------|------|------|------|
| | **GA** | **WOA** | **AOA** | **GA** | **WOA** | **AOA** |
| 1 | 21,724,085.81 | 21,551,287.45 | 22,142,283.58 | 28.34 | 25.34 | 26.51 |
| 2 | 25,674,485.33 | 25,661,665.47 | 26,087,638.32 | 37.48 | 32.76 | 34.60 |
| 3 | 28,648,512.24 | 28,945,913.59 | 29,157,515.63 | 48.90 | 45.18 | 46.63 |
| 4 | 32,649,844.27 | 32,611,180.57 | 33,294,422.55 | 61.91 | 54.67 | 57.49 |
| 5 | 35,997,466.24 | 35,498,153.74 | 36,576,978.90 | 80.17 | 73.21 | 75.92 |
| 6 | 39,745,648.11 | 38,977,673.10 | 39,062,457.88 | 102.34 | 93.94 | 97.22 |
| 7 | 44,587,444.51 | 45,195,047.80 | 44,282,226.46 | 127.94 | 117.98 | 121.86 |
| 8 | 48,679,748.36 | 48,604,112.29 | 47,925,596.33 | 166.76 | 152.64 | 158.15 |
| 9 | 53,354,731.64 | 53,351,761.01 | 52,351,352.83 | 218.51 | 200.67 | 207.63 |
| 10 | 57,884,622.47 | 58,280,035.03 | 57,288,049.32 | 289.79 | 267.94 | 276.46 |
| Mean | 38,894,658.90 | 38,867,683.00 | 38,816,852.18 | 116.21 | 106.43 | 110.25 |

The results of numerical examples of larger sizes show that, on average, the AOA obtained the lowest average of the total OBF of the problem in 10 numerical examples. However, the WOA was able to achieve a near-optimal solution in a shorter computing time than the GA and AOA. Figure 6 also shows the average results obtained from meta-heuristic algorithms regarding the OBF and computational time.

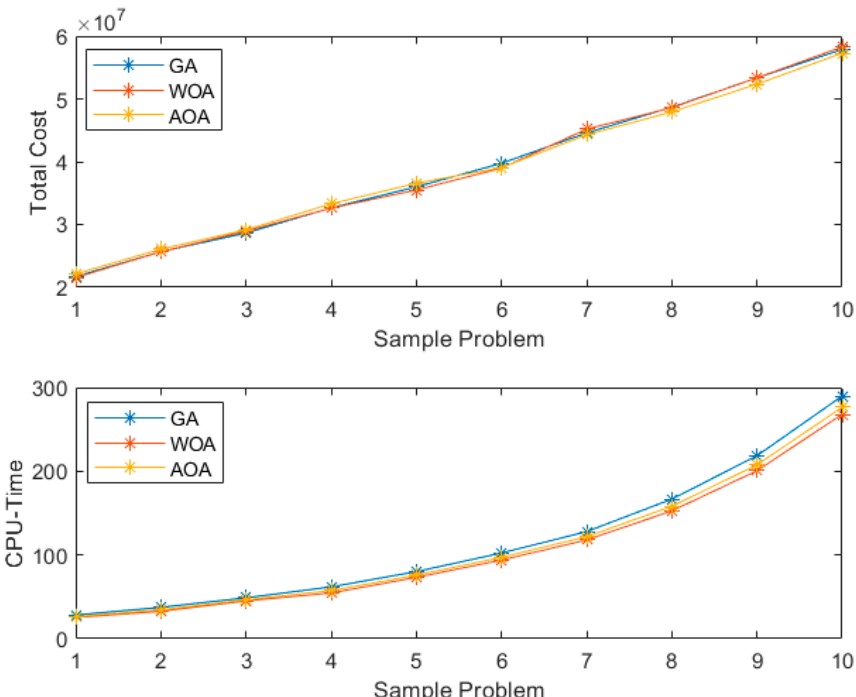

**Figure 6.** Average OBF and computational time in numerical examples of larger sizes with meta-heuristic algorithms.

The analyses demonstrate that the AOA and the WOA are efficient at finding near-optimal solutions and solving numerical cases quickly. Thus, the *t*-test statistical test was applied with 95% confidence to evaluate the data. The numerical example averages alter significantly if the *p*-value is less than 0.05. Table 9 provides the *t*-test statistical test results for OBF and computation time metrics.

**Table 9.** The results of the *t*-test statistical test to check the averages of the OBF and computing time.

| Index | Algorithm | Estimate for Difference | 95% CI for Difference | *t*-Value | *p*-Value |
|-------|-----------|------------------------|----------------------|-----------|-----------|
| | GA-WOA | 26,976 | $(-10{,}275{,}601, 10{,}329{,}553)$ | 0.01 | 0.996 |
| Total Cost | GA-AOA | 77,807 | $(-9{,}953{,}491, 10{,}109{,}104)$ | 0.02 | 0.987 |
| | WOA-AOA | 50,831 | $(-10{,}028{,}193, 10{,}129{,}854)$ | 0.01 | 0.992 |
| | GA-WOA | 9.8 | $(-60.2, 79.8)$ | 0.29 | 0.773 |
| CPU-Time | GA-AOA | 6.0 | $(-65.0, 77.0)$ | 0.18 | 0.862 |
| | WOA-AOA | 3.8 | $(-64.6, 72.2)$ | 0.12 | 0.908 |

Table 9 shows that all of the meta-heuristic methods express average outcomes with *p*-values greater than 0.05.

Thus, the AOA solves the distribution network model of perishable agricultural items better than the other algorithms. The *t*-test was used to evaluate the level of convergence, or whether or not the average of the sample is the same as the average of the population in a situation where the standard deviation of the population is unknown. Because the t-distribution in the case of small samples is adjusted using degrees of freedom, it is possible to use this test for very small samples. The results of the statistical analysis show that the WOA method has a lower runtime, and, as a result, it is a more efficient method.

## 5. Conclusions

Achieving food security is one of the basic requirements of societies. Considering the importance of the growth and development of the agricultural sector in achieving food security, it is very important to adopt and implement policies and measures that lead to increasing the productivity of the agricultural sector. Creating a supply chain for basic agricultural products by including a wide range of agricultural sub-sectors from the supply of required production inputs to the final consumer's access to the goods will lead to an increase in the productivity of the agricultural sector, as well as the upstream and downstream industries related to it. Therefore, taking into account the importance of creating an optimal supply chain for various agricultural products, in this study, the situation of the distribution supply chain of agricultural products has been examined. Uncertainties have also been considered in this study, until the feasibility of the problem was not lost. In this paper, we looked at how to simulate highly perishable agricultural goods when the demand from merchants and the transportation costs are unknown. This research presents a model for optimizing the distribution systems of agricultural products with high perishability in an innovative way. The presented model seeks to optimize the amount of economic order, the allocation of distribution network levels, the routing of distribution items to retailers, and the management of retailers' warehouses and distribution centers in order to minimize the costs of the entire distribution network of agricultural products. The RPO technique was used to regulate the non-crisp parameters, and the analysis of the numerical example demonstrated that rising uncertainty rates led to rising overall network costs as a consequence of rising potential demand. The rising price tag may be attributed to higher order processing, warehousing, and delivery expenses. The high rate of perishability of agricultural goods has also been seen to raise the expenses associated with it, such as the costs of product failure, and, hence, the costs of the agricultural distribution network. In addition, the impact of the mathematical model's robustness on the management of non-crisp parameters was investigated, and it was found that, as robustness grew, the overall network costs rose, but the standard deviation fell. This demonstrates the RPO method's efficacy in forecasting merchants' unpredictable demand. Furthermore, the outcomes of solving bigger numerical cases using GA, WOA, and AOA revealed that the AOA is more efficient in finding the best value of the OBF. While the other techniques took longer, the WOA solved the mathematical model in record speed. In addition, the statistical examination of the *t*-test showed that the averages produced from the OBF and the computing time across all of the solution approaches are not significantly different. In order to reduce the negative effect of the high perishability of goods in agricultural products, or fast-consumption products, it is suggested to provide a mathematical model to improve the quality of the agricultural products that spoil quickly and simultaneously reduce production costs. Both innovative approaches to problem solving and approximation techniques are supported. Considering the high level of uncertainty (with the conditionability of the problem) can increase the quality of the mathematical model. The high level of complexity of the model and the failure to reach a reasonable solution if the number of uncertainties increases are the basic limitations of this model. Due to the fact that the lack of appropriate policies regarding the management of the supply chain of agricultural products can lead to the inefficiency of the supply chain of agricultural products in various stages (including the supply of inputs and the production, distribution, and marketing of products), it is suggested that related planning and policies be carried out in order to improve the infrastructure, which leads to increasing the efficiency of some logistics subsystems, such as transportation, storage, communication, and information systems, etc.

**Author Contributions:** A.D.: Conceptualization, Methodology, and Writing—review and editing; R.R.: Data curation and Writing—review and editing; P.G.: Project administration, Formal analysis, Supervision, and Visualization; M.B.: Validation, Software, and Investigation; A.P.C.: Formal analysis, Writing—original draft, and Writing—review and editing. All authors have read and agreed to the published version of the manuscript.

**Funding:** This research received no external funding.

**Institutional Review Board Statement:** Not applicable.

**Informed Consent Statement:** Not applicable.

**Data Availability Statement:** Data available on request due to restrictions e.g., privacy or ethical.

**Conflicts of Interest:** The authors declare no conflict of interest.

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
