# Peer review of "Design of an Optimal Robust Possibilistic Model in the Distribution Chain Network of Agricultural Products with High Perishability under Uncertainty"

_sustainability, doi:10.3390/su151511669_

Round 1

Reviewer 1 Report

This manuscript discussed the model of a distribution network problem of highly perishable agricultural products under uncertainty which has strong practical significance.

1. Formats need a lot of improvement.

2. "Literature review" should be more logical.

3. The theoretical value, application value and innovation of the manuscript should be more clearly defined in the conclusion.

4. If you have an application case, please add it to the manuscript.

Author Response

Reviewer 1

This manuscript discussed the model of a distribution network problem of highly perishable agricultural products under uncertainty which has strong practical significance.

  1. Formats need a lot of improvement.

Thanks to the reviewer, the authors tried to modify the format as much as possible.

  1. "Literature review" should be more logical.

Thanks to the reviewer, sections were added to the literature review.

  1. The theoretical value, application value and innovation of the manuscript should be more clearly defined in the conclusion.

Thanks to the reviewer, this section was added to the abstract and introduction.

  1. If you have an application case, please add it to the manuscript.

Thanks to the respected reviewer, there is no case study in this research.

Reviewer 2 Report

This paper formulated the distribution network model of agricultural products with high perishability under uncertainty to minimize the total cost. The genetic algorithm (GA), Whale Optimization Algorithm (WOA), and Arithmetic optimization algorithm (AOA) are applied to solve the proposed model for large cases. The proposed model is interesting and the structure of this article is sufficient. There are some revisions as follows:

- The authors needed to describe the scientific contribution better.

- Literature review; The author should clearly explain the difference between this paper and the existing literature.

- Page 5, Line 240: Please recheck the section number.

- Please write each algorithm's full name before using abbreviations (GA, WOA, and AOA).

- Should the paragraphs on lines 345-348 be under the heading RPO approach?

- Figure 3: The numbers of retailers and distribution Centers in Figure 3 need to be revised to be clearer.

- The experimental results showed that the total cost obtained by the three methods was slightly different, and the statistical results showed no significant difference. I wonder why the authors conclude that AOA is the most efficient algorithm compared to other algorithms.

- The authors need to explain the statistical results in more detail.

- Conclusions; It should mention the future steps to cover the weak points of the current method.

fine

Author Response

Reviewer 2

This paper formulated the distribution network model of agricultural products with high perishability under uncertainty to minimize the total cost. The genetic algorithm (GA), Whale Optimization Algorithm (WOA), and Arithmetic optimization algorithm (AOA) are applied to solve the proposed model for large cases. The proposed model is interesting and the structure of this article is sufficient. There are some revisions as follows:

- The authors needed to describe the scientific contribution better.

Thanks to the reviewer, this section was added to the abstract and introduction.

- Literature review; The author should clearly explain the difference between this paper and the existing literature.

Thanks to the reviewer, sections were added to the literature review.

- Page 5, Line 240: Please recheck the section number.

With thanks and appreciation to the respected reviewer, the review and correction was done.

- Please write each algorithm's full name before using abbreviations (GA, WOA, and AOA).

With thanks and appreciation to the respected reviewer, the review and correction was done.

- Should the paragraphs on lines 345-348 be under the heading RPO approach?

Thanks to the reviewer, yes it should be in this section.

- Figure 3: The numbers of retailers and distribution Centers in Figure 3 need to be revised to be clearer.

With thanks and appreciation to the respected reviewer, the review and correction was done.

- The experimental results showed that the total cost obtained by the three methods was slightly different, and the statistical results showed no significant difference. I wonder why the authors conclude that AOA is the most efficient algorithm compared to other algorithms.

With gratitude to the respected reviewer, the calculation results show the speed and lower number of repetitions in this algorithm and therefore this algorithm is more efficient.

- The authors need to explain the statistical results in more detail.

With thanks and appreciation to the respected reviewer, the review and correction was done.

 - Conclusions; It should mention the future steps to cover the weak points of the current method.

Thanks to the reviewer, this section was added to the conclusion.

Reviewer 3 Report

This study presents the framework of a distribution network for agricultural products, with particular attention to products with rapid and high perishability from the time they are harvested until they reach the final consumer.

Below are my comments that may help the authors further improve their manuscript:

1. Line 28: Please write the full definition for these abbreviations, at least for the first time (GA, WOA, and AOA). Furthermore, add the full definition for all abbreviations throughout the manuscript’s text for the first time. Or add the list of abbreviations or nomenclature at the beginning.

2. Line 331: Replace Equation with Equations (10) and (11) calculate ……..

3.  Line 397: Please revise this sentence.

4. Figure 3: The numbers inside the ovals, triangles, and rectangles are unclear. Please amend them.

5. In the conclusion section: The authors should mention the limitations of this study and add their future perspectives for this paper. Please add these to the manuscript’s text.

6. In the references section: The authors should follow the journal format and style in writing this section according to the journal’s guidelines. Please revise all references one by one and amend them according to the journal’s format and style.

7. Line 640: The reference number is 5, not 15; please revise and amend it.

Author Response

Reviewer 3

This study presents the framework of a distribution network for agricultural products, with particular attention to products with rapid and high perishability from the time they are harvested until they reach the final consumer.

Below are my comments that may help the authors further improve their manuscript:

  1. Line 28: Please write the full definition for these abbreviations, at least for the first time (GA, WOA, and AOA). Furthermore, add the full definition for all abbreviations throughout the manuscript’s text for the first time. Or add the list of abbreviations or nomenclature at the beginning.

With thanks and appreciation to the respected reviewer, the review and correction was done.

  1. Line 331: Replace Equation with Equations (10) and (11) calculate ……..

With thanks and appreciation to the respected reviewer, the review and correction was done.

  1. Line 397: Please revise this sentence.

With thanks and appreciation to the respected reviewer, the review and correction was done.

  1. Figure 3: The numbers inside the ovals, triangles, and rectangles are unclear. Please amend them.

With thanks and appreciation to the respected reviewer, the review and correction was done.

  1. In the conclusion section: The authors should mention the limitations of this study and add their future perspectives for this paper. Please add these to the manuscript’s text.

With thanks and appreciation to the respected reviewer, the review and correction was done.

  1. In the references section: The authors should follow the journal format and style in writing this section according to the journal’s guidelines. Please revise all references one by one and amend them according to the journal’s format and style.
  2. Line 640: The reference number is 5, not 15; please revise and amend it.

With thanks and appreciation to the respected reviewer, the review and correction was done.

Round 2

Reviewer 1 Report

Authors did not revise the paper carefully according to my suggestions, especially the literature review and conclusion, which are very confusing.

Author Response

The comments were checked again and your suggestions were added to the article and modified.

Thank you very much for your attention.

Reviewer 3 Report

The authors have responded to my comments point by point, and I am satisfied with their responses.

Author Response

Thank you very much.